# The Isolation, Structure Elucidation and Bioactivity Study of Chilensosides A, A_1_, B, C, and D, Holostane Triterpene Di-, Tri- and Tetrasulfated Pentaosides from the Sea Cucumber *Paracaudina chilensis* (Caudinidae, Molpadida)

**DOI:** 10.3390/molecules27217655

**Published:** 2022-11-07

**Authors:** Alexandra S. Silchenko, Sergey A. Avilov, Pelageya V. Andrijaschenko, Roman S. Popov, Ekaterina A. Chingizova, Boris B. Grebnev, Anton B. Rasin, Vladimir I. Kalinin

**Affiliations:** G.B. Elyakov Pacific Institute of Bioorganic Chemistry, Far Eastern Branch of the Russian Academy of Sciences, Pr. 100-letya Vladivostoka 159, 690022 Vladivostok, Russia

**Keywords:** *Paracaudina chilensis*, Molpadida, triterpene glycosides, chilensosides, sea cucumber, cytotoxic activity

## Abstract

Five new triterpene (4,4,14-trimethylsterol) di-, tri- and tetrasulfated pentaosides, chilensosides A (**1**), A_1_ (**2**), B (**3**), C (**4**), and D (**5**) were isolated from the Far-Eastern sea cucumber *Paracaudina chilensis*. The structures were established on the basis of extensive analysis of 1D and 2D NMR spectra and confirmed by HR-ESI-MS data. The structural variability of the glycosides concerned the pentasaccharide chains. Their architecture was characterized by the upper semi-chain consisting of three sugar units and the bottom semi-chain of two sugars. Carbohydrate chains of compounds **2**–**5** differed in the quantity and positions of sulfate groups. The interesting structural features of the glycosides were: the presence of two sulfate groups at C-4 and C-6 of the same glucose residue in the upper semi-chain of **1**, **2**, **4**, and **5** and the sulfation at C-3 of terminal glucose residue in the bottom semi-chain of **4** that makes its further elongation impossible. Chilensoside D (**5**) was the sixth tetrasulfated glycoside found in sea cucumbers. The architecture of the sugar chains of chilensosides A–D (**1**–**5**), the positions of sulfation, the quantity of sulfate groups, as well as the aglycone structures, demonstrate their similarity to the glycosides of the representatives of the order Dendrochirotida, confirming the phylogenetic closeness of the orders Molpadida and Dendrochirotida. The cytotoxic activities of the compounds **1**–**5** against human erythrocytes and some cancer cell lines are presented. Disulfated chilensosides A_1_ (**2**) and B (**3**) and trisulfated chilensoside C (**4**) showed significant cytotoxic activity against human cancer cells.

## 1. Introduction

Despite triterpene glycosides from sea cucumbers having a rather long history of investigations, there are some systematic groups, including the order Molpadida, comprising the studied species *Paracaudina chilensis*, which are poorly studied or unexplored chemically. The majority of recent research concerning the sea cucumber triterpene glycosides deals with the structure elucidation of the compounds isolated from representatives of the orders Dendrochirotida, Synallactida, and Holothuriida [1,2,3,4,5,6,7,8,9]. The use of mass-spectrometry-based metabolomics for the solving of diverse chemical and biological issues concerning secondary metabolites has become very popular and has provided some significant results in the exploration of triterpene glycoside chemical diversity, their content and composition in different body parts [10,11,12,13], and their chemotaxonomy [14,15]. The application of this approach in combination with molecular phylogenetic analysis allowed to clarify the evolution of the Holothuroidea taxons [16]. Different investigations have also confirmed the defensive role of glycosides [17,18]. The biosynthetic studies of triterpene glycosides are very difficult to conduct. They began in the 1970s through the introduction of radioactively labeled precursors to the organisms-producers and showed some contradictory results [19,20,21]. However, it has been established that precursors of triterpenoids can be either lanosterol or parkeol depending on the intranuclear double bond position in the biosynthesizing aglycones [22,23,24,25]. Oxydosqualenecyclases (OSCs) are the enzymes processing the cyclization of 2,3-oxidosqualene, making diverse triterpene alcohols. This stage is the branchpoint of steroids and triterpenoids biosynthesis. In the process of steroidogenesis, the triterpene precursors are enzymatically demethylated at positions C-4, C-14, double bond positions are changed and side chains are modified. The sea cucumbers are characterized by the presence of steroids with uncommon chemical features (Δ^9(11)^- and Δ^7^-derivatives) instead of Δ^5^-sterols characteristic for other animals. This is explained by the presence of the membranolytic triterpene glycosides targeting the Δ^5^-steroids. Therefore, to protect their own membranes from the action of these toxins, the steroid composition has been evolutionary changed. It was supposed that Δ^7^-steroids are formed as result of modification of dietary Δ^5^-sterols, while Δ^9(11)^-steroid compounds are biosynthesized from parkeol, formed *de novo* by OSCs in the sea cucumbers. Noticeably, some sea cucumber species contain 14α-methylated and 4α,14α-dimethylated Δ^9(11)^-sterols [23]. The recent genetic studies of *Apostichopus japonicus* explained this phenomenon by the absence of the gene for 14-sterol-demethylase in its genome [26]. There is little molecular/genetic research on genes and corresponding enzymes participating in triterpenoid biosynthesis in sea cucumbers. They only concerned transcriptomic analysis of tissues of *Holothuria scabra* and *Stichopus horrens* and have led to identification of the sequences corresponding to some genes of the mevalonate pathway [27,28]. The decoding of sequences of OSCs from *Apostichopus japonicus* followed by the expression of the genes integrated into the yeast genome allowed for the identification of parkeol and 9βН-lanosta-7,24-dien-3*β*-ol as the products [26]. 

Finally, there exists a large number of investigations devoted to the biological activity, including anticancer activity, of sea cucumber glycosides, which are valuable sources of new drug candidates [29,30,31,32,33,34,35].

Only preliminary research on the glycosidic composition of *Paracaudina ransonetii* (=*Paracaudina chilensis*) has been previously published [36]. The taxonomic status of the order Molpadida raises questions for biologists dealing with the systematics of Holothuroidea. Some support the idea of closeness of Molpadida and Dendrochirotida, while others consider molpadiids as being near to Aspidochirotida [37]. From this viewpoint, investigations on the glycosides of representatives of the order Molpadida are relevant for the searching for new structural variants, which broaden our knowledge concerning chemical biodiversity and chemotaxonomy. Glycosides have been successfully used as chemotaxonomic markers of different sea cucumber systematic groups [38,39,40,41]. Therefore, the analysis of chemical peculiarities of the glycosides of *P. chilensis* can help to resolve this dilemma.

New triterpene glycosides, chilensosides A (**1**), A_1_ (**2**), B (**3**), C (**4**) and D (**5**), were isolated from the Far Eastern sea cucumber *Paracaudina chilensis*. The chemical structures of **1**–**5** were established by the analyses of the ^1^H, ^13^C NMR, 1D TOCSY and 2D NMR (^1^H, ^1^H-COSY, HMBC, HSQC, ROESY) spectra as well as HR-ESI mass spectra. All the original spectra are presented in Appendix A. The hemolytic activity against human erythrocytes and cytotoxic activities against human neuroblastoma SH-SY5Y, adenocarcinoma HeLa, colorectal adenocarcinoma DLD-1, leukemia promyeloblast HL-60 and monocytic THP-1 cells were examined.

## 2. Results and Discussion

### 2.1. Structural Elucidation of the Glycosides

The crude glycosidic fraction of the sea cucumber *Paracaudina chilensis* was obtained as a result of hydrophobic chromatography of the concentrated ethanolic extract on a Polychrom-1 column (powdered Teflon, Biolar, Latvia). Its subsequent separation by chromatography on Si gel columns with the stepped gradient of the system of eluents CHCl3/EtOH/H2O used in ratios (100:100:17), (100:125:25), and (100:150:50) gave the fractions I–III. Each of the obtained fractions was additionally purified on a Si gel column with the solvent system CHCl3/EtOH/H2O (100:125:25), which resulted in the isolation of five subfractions I.0, I.1, II, III.1 and III.2. The individual compounds **1**–**5** (Figure 1) were isolated through HPLC of these subfractions on the silica-based column Supelcosil LC-Si (4.6 *×* 150 mm), and reversed-phase columns Supelco Discovery HS F5-5 (10 *×* 250 mm) and Diasfer 110 C-8 (4.6 × 250 mm).

The configurations of the monosaccharide residues in the glycosides **1**–**5** were assigned as *D* based on the biogenetic analogies with the monosaccharides from all other known sea cucumber triterpene glycosides.

The molecular formula of chilensoside A (**1**) was determined to be C_60_H_92_O_34_S_2_Na_2_ from the [M_2Na_–Na]^−^ ion peak at *m/z* 1443.4800 (calc. 1443.4815), and [M_2Na_–2Na]^2−^ ion peak at *m/z* 710.2466 (calc. 710.2461) in the (*−*) HR-ESI-MS (Appendix A). The ^13^C NMR spectrum of the aglycone part of chilensoside A (**1**) demonstrated the signals of quaternary oxygen-bearing carbons at δ_C_ 176.8 (C-18) and 82.8 (C-20), corresponding to 18(20)-lactone and the signals of olefinic carbons at δ_C_ 151.1 (C-9), 111.2 (C-11) (Table 1, Appendix A), indicating the presence of 9(11)-double bond, typical of many sea cucumber glycosides. An additional deshielded signal at δ_C_ 214.6 was assigned to C-16 oxo-group in the holostane nucleus, confirmed by the singlet signal of H-17 at δ_H_ 2.89 with a corresponding carbon signal at δ_C_ 61.2 (C-17). The protons of the side chain H-22/H-23/H-24 formed the isolated spin system deduced by the COSY spectrum, indicating the presence of an additional 23*Z*(24)-double bond (δ_H-23_ 5.90 (dd, *J* = 6.3; 11.8 Hz), δ_H-24_ 5.90 (d, *J* = 11.8 Hz)). The presence of the signal of quaternary oxygen-bearing carbon at δ_C_ 81.3 along with the coincidence of the signals of 26, 27-methyl groups to each other (δ_C_ 24.7 (C-26, C-27), δ_H_ 1.42 (s, H-26, H-27)) indicated the presence of hydroxyl at C-25. The side chain structure was confirmed by the HMBC correlations H-24/C-25; H-23/C-22; H-26(27)/C-24 (Table 1). The same aglycone was found only once earlier in cladoloside A_5_ from the sea cucumber *Cladolabes schmeltzii* [42].

The ^1^H and ^13^C NMR spectra of the carbohydrate chain of chilensosides A (**1**) (Table 2, Appendix A) and A_1_ (**2**) (Appendix A, Appendix A) were coincident to each other, indicating the identity of sugar moieties of **1**, **2**. The spectra of **1** demonstrated five characteristic doublets of anomeric protons at δ_H_ 4.66–5.20 (*J* = 7.1–8.1 Hz) and the signals of anomeric carbons at δ_C_ 102.2–104.7, indicating the presence of a pentasaccharide chain and *β*-configurations of glycosidic bonds. The coherent analysis of the ^1^H, ^1^H-COSY, 1D TOCSY, HSQC and ROESY spectra of **1** indicated the presence of one xylose (Xyl1), one quinovose (Qui2), two glucose (Glc3 and Glc4), and 3-*O*-methylglucose (MeGlc5) residues. The ROE- and HMBC correlations showed the positions of glycosidic linkages (Table 2, Appendix A), which indicated that the carbohydrate chain of **1** was branched by C-4 Xyl1 having bottom semi-chain composed of two sugar units and the upper semi-chain from three.

Noticeably, such architecture of carbohydrate chains is not common for the holothuroid glycosides, but similar sugar moieties have been found in some glycosides of recently studied species of sea cucumbers: *Thyonidium kurilensis* [43] and *Psolus chitonoides* [6] (order Dendrochirotida).

The availability of two sulfate groups in the sugar moiety of **1** was deduced on the basis of shifting effects observed in its ^13^C NMR spectrum. These were the signals of two hydroxy methylene groups of glucopyranose residues at δ_C_ 61.8 (C-6 Glc3) and 62.0 (C-6 MeGlc5), indicating the absence of sulfate groups in these positions and one signal at δ_C_ 68.2 (deshielded due to α-shifting effect of sulfate group) corresponding to sulfated at C-6 Glc4 residue. Additional shifting effects of the sulfate group became evident when the ^13^C NMR spectrum of **1** was compared with the spectrum of the carbohydrate part of kuriloside A_1_ [43]. The signals of all monosaccharides in the spectra of these glycosides were close to each other, with the exception of the signals of glucose residue in the upper semi-chain (Glc4). The signals of C-3 Glc4 and C-5 Glc4 were shielded in the spectrum of **1** (to δ_C_ 82.9 and 74.5, correspondingly) in comparison with the same signals in the spectrum of kuriloside A_1_ (δ_C_ 86.9 and 75.7, correspondingly) due to the *β*-shifting effect of the sulfate group, which was attached to C-4 Glc4 of **1**. This was confirmed by an α-shifting effect: the signal of C-4 Glc4 in the spectrum of chilensoside A (**1**) was deshielded (δ_C_ 75.6) when compared with the signal of C-4 Glc4 in the spectrum of kuriloside A_1_ (δ_C_ 69.6). Therefore, two sulfate groups were attached to one monosaccharide unit (Glc4) in the sugar chain of **1**. Such a structural feature was also recently found in psolusoside P from *Psolus fabricii* [44]. However, chilensoside A (**1**) is a new combination of some unusual structural features: aglycone side chain structure, carbohydrate chain architecture and the positions of sulfate groups. 

The (*−*)ESI-MS/MS of **1** (Appendix A) demonstrated the fragmentation of [M_2Na_−Na]^−^ ion at *m/z* 1443.5 with ion peaks observed at *m/z* 1179.5 [M_2Na_−Na−Glc–SO_3_Na+2H]^−^, 1135.5 [M_2Na_−Na−Glc–Qui+H]^−^, 1010.4 [M_2Na_−Na−MeGlc−2HSO_4_Na]^−^, 417.1 [M_2Na_−Na−MeGlc−Glc(OSO_3_Na)_2_−Agl]^−^, and 255.0 [M_2Na_−Na−MeGlc−Glc(OSO_3_Na)_2_−Glc−Agl]^−^, corroborating the sequence of monosaccharides and the aglycone structure of **1**.

These data indicate that chilensoside A (**1**) is 3*β*-*O*-{*β*-D-glucopyranosyl-(1 → 4)-*β*-D-quinovopyranosyl-(1 → 2)-[3-*O*-methyl-*β*-D-glucopyranosyl-(1 → 3)-4,6-*O*-sodium disulfate-*β*-D-glucopyranosyl-(1 → 4)]-*β*-D-xylopyranosyl}-16-oxo,25-hydroxyholosta-9(11),23*Z*(24)-diene.

The aglycones of chilensosides A_1_ (**2**), B (**3**), C (**4**) and D (**5**) (Table 3, Appendix A) were identical to each other and to those of cladoloside A_4_ [42] and psolusoside D_1_ [45]. This holostane aglycone has the same polycyclic system as **1** and differs in the side chain structure with a 24(25)-double bond.

The molecular formula of chilensoside A_1_ (**2**) was determined to be C_60_H_92_O_33_S_2_Na_2_ from the [M_2Na_−Na]^−^ ion peak at *m/z* 1427.4928 (calc. 1427.4865), and [M_2Na_–2Na]^2−^ ion peak at *m/z* 702.2510 (calc. 702.2487) in the (−)HR-ESI-MS (Appendix A). The (*−*)ESI-MS/MS of **2** (Appendix A) demonstrated the fragmentation of [M_2Na_−Na]^−^ ion at *m/z* 1427.5, *m/z*: 1120.5 [M_2Na_−Na−Glc−Qui+H]^−^, 915.4 [M_2Na_−Na−Glc−Qui−2SO_3_Na+3H]^−^, 667.1 [M_2Na_−Na−Glc−Qui–Agl]^−^, 417.1 [M_2Na_−Na−MeGlc−Glc(OSO_3_Na)_2_−Agl]^−^.

All these data indicate that chilensoside A_1_ (**2**) is 3*β*-*O*-{*β*-D-glucopyranosyl-(1 → 4)-*β*-D-quinovopyranosyl-(1 → 2)-[3-*O*-methyl-*β*-D-glucopyranosyl-(1 → 3)-4,6-*O*-sodium disulfate-*β*-D-glucopyranosyl-(1 → 4)]-*β*-D-xylopyranosyl}-16-oxoholosta-9(11),24(25)-diene.

The molecular formula of chilensoside B (**3**) was determined to be C_60_H_92_O_33_S_2_Na_2_ from the [M_2Na_–Na]^−^ ion peak at *m/z* 1427.4881 (calc. 1427.4865), and [M_2Na_–2Na]^2−^ ion peak at *m/z* 702.2499 (calc. 702.2487) in the (*−*)HR-ESI-MS (Appendix A). The ^1^H and ^13^C NMR spectra of the carbohydrate chain of chilensoside B (**3**) (Table 4, Appendix A) demonstrated five characteristic doublets of anomeric protons at δ_H_ 4.66–5.18 (*J* = 7.1–8.1 Hz) and five signals of anomeric carbons at δ_C_ 102.3–104.7, indicating the presence of a pentasaccharide chain and *β*-configurations of glycosidic bonds. The extensive analysis of the ^1^H, ^1^H-COSY, 1D TOCSY, HSQC, ROESY and HMBC spectra (Table 4, Appendix A) of **3** indicated the same monosaccharide composition, positions of glycosidic linkages, and architecture established for the glycosides **1**, **2**. The differences in the chemical shifts of carbon signals of chilensosides A (**1**) and B (**3**) were attributed to the diverse positions of sulfate groups. The signal of C-4 Glc4 in the ^13^C NMR spectrum of **3** was shielded to δ_C_ 68.9 instead of δ_C_ 75.6 in **1** due to the absence of a sulfate group in this position of **3**. Additionally, the signal of C-3 Glc4 was deshielded to 85.9 due to the glycosylation effect and the absence of the *β*-shifting effect of sulfate group. The signal of C-6 Glc4 at δ_C_ 67.2 was characteristic for the sulfated hydroxy methylene group of the glucopyranose unit. Therefore, the glucose residue attached to C-4 Xyl1 of the carbohydrate chain of **3** bears one sulfate group at C-6. The comparison of the signals assigned to carbons of the 3-*O*-methylglucose unit of the compounds **3** (Table 4) and **1** (Table 2) showed that the signal of C-4 MeGlc5 of **3** was deshielded by 6.1 ppm (to δ_C_ 76.1) and the signals of C-3 and C-5 MeGlc5 were shielded by 1.7 and 1.0 ppm, corresponding to the shifting effects of the sulfate group attached to C-4 MeGlc5 of chilensoside B (**3**). Thus, the glycoside **3** is a new disulfated pentaoside having sulfate groups at C-6 Glc4 and C-4 MeGlc5. The compound with identical positions of sulfates but differing in the terminal xylose residue in the bottom semi-chain was chitonoidoside H, found recently in the sea cucumber *Psolus chitonoides* [6]. 

The (*−*)ESI-MS/MS of **3** (Appendix A) demonstrated the fragmentation of [M_2Na_−Na]^−^ ion at *m/z* 1427.5, resulting in the ion peaks appearance at *m/z* 1307.5 [M_2Na_−Na−NaHSO_4_]^−^, 1149.5 [M_2Na_−Na−MeGlcOSO_3_Na+H]^−^, 987.4 [M_2Na_−Na−MeGlcOSO_3_Na−Glc+H]^−^, 841.4 [M_2Na_−Na−MeGlcOSO_3_Na−Glc−Qui+H]^−^, 667.1 [M_2Na_−Na−Agl−Glc−Qui−H]^−^. The fragmentation of [M_2Na_−2Na]^2−^ ion at *m/z* 702.2 led to the ion peak at *m/z* 621.7 [M_2Na_−2Na−Glc]^2−^, and 548.2 [M_2Na_−2Na−Glc−Qui]^2−^, confirming the structure of **3**.

These data indicate that chilensoside B (**3**) is 3*β*-*O*-{*β*-D-glucopyranosyl-(1 → 4)-*β*-D-quinovopyranosyl-(1 → 2)-[4-*O*-sodium sulfate-3-*O*-methyl-*β*-D-glucopyranosyl-(1 → 3)-6-*O*-sodium sulfate-*β*-D-glucopyranosyl-(1 → 4)]-*β*-D-xylopyranosyl}-16-oxoholosta-9(11),24(25)-diene.

The molecular formula of chilensoside C (**4**) was determined to be C_60_H_91_O_36_S_3_Na_3_ from the [M_3Na_–Na]^−^ ion peak at *m/z* 1529.4300 (calc. 1529.4253), [M_3Na_–2Na]^2−^ ion peak at *m/z* 753.2206 (calc. 753.2180) and [M_3Na_–3Na]^3−^ ion peak at *m/z* 494.4839 (calc. 494.4823) in the (*−*)HR-ESI-MS (Appendix A).

The ^1^H and ^13^C NMR spectra of the carbohydrate chain of chilensoside С (**4**) (Table 5, Appendix A) demonstrated five characteristic doublets of anomeric protons at δ_H_ 4.65–5.21 (*J* = 6.5–8.5 Hz) and five signals of anomeric carbons at δ_C_ 102.4–104.7, indicating the presence of a pentasaccharide chain and *β*-configurations of glycosidic bonds. 

The extensive analysis of the ^1^H, ^1^H-COSY, 1D TOCSY, HSQC, ROESY, and HMBC spectra of **4** indicated the same monosaccharide composition, glycosidic bond locations and architecture of carbohydrate chains as in the previously discussed glycosides **1**−**3**. Differences were found in the quantity of sulfate groups, which was also confirmed by MS data, where three-charged ions were registered, indicating the presence of three sulfate groups. 

The comparison of the ^13^C NMR spectra of sugar moieties of **4** and **1** showed the coincidence of all the signals except the signals of glucose residue in the bottom semi-chain. The signal of C-3 Glc3 was deshielded to δ_C_ 84.3 in the spectrum of **4**, which could be explained by the *α*-shifting effect of the sulfate group as well as by the glycosylation effect. However, the latter was excluded due to the absence of the ROE- and HMBC correlations of H-3 Glc3 with any protons or carbons of neighboring monosaccharide residues (Table 5). Moreover, the signals of C-2 Glc3 and C-4 Glc3 in the spectrum of **4** were shielded to δ_C_ 73.1 and 69.8, respectively, in comparison with the corresponding signals in the spectrum of **1** due to *β*-shifting effect of sulfate group at C-3 Glc3. Therefore, the third sulfate group in chilensoside C (**4**) was unique for the glycosides position at C-3 Glc3 instead of the characteristic glycosidic bond position in the glycosides with normal (consisting of three monosaccharide units) bottom semi-chain. Such a location of the sulfate group makes further elongation of the carbohydrate chain of **4** impossible. The rest of the sulfate groups were attached to C-4 Glc4 and C-6 Glc4 in chilensoside C (**4**), by the same manner as in chilensosides A (**1**), and A_1_ (**2**). 

The (*−*)ESI-MS/MS of **4** (Appendix A) demonstrated the fragmentation of [M_3Na_−Na]^−^ ion at *m/z* 1529.5, which resulted in the ion peaks at *m/z* 1015.4 [M_3Na_−Na−GlcOSO_3_Na−Qui−SO_3_Na]^−^, 987.4 [M_3Na_−Na−MeGlc−Glc(OSO_3_Na)_2_]^−^, 605.2 [M_3Na_−Na−MeGlc−NaHSO_4_]^−^. The fragmentation of [M_3Na_−2Na]^2−^ ion at *m/z* 753.2 led to the presence of the ion peaks at *m/z* 702.2 [M_3Na_−2Na−SO_3_Na]^2−^, 605.2 [M_3Na_−2Na−MeGlc−NaHSO_4_]^2−^.

These data indicate that chilensoside C (**4**) is 3*β*-*O*-{3-*O*-sodium sulfate-*β*-D-glucopyranosyl-(1 → 4)-*β*-D-quinovopyranosyl-(1 → 2)-[3-*O*-methyl-*β*-D-glucopyranosyl-(1 → 3)-4,6-*O*-sodium disulfate-*β*-D-glucopyranosyl-(1 → 4)]-*β*-D-xylopyranosyl}-16-oxoholosta-9(11),24(25)-diene.

The molecular formula of chilensoside D (**5**) was determined to be C_60_H_90_O_39_S_4_Na_4_ from the [M_4Na_–Na]^−^ ion peak at *m/z* 1631.3667 (calc. 1631.3641), [M_4Na_–2Na]^2−^ ion peak at *m/z* 804.1886 (calc. 804.1874), [M_4Na_–3Na]^3−^ ion peak at *m/z* 528.4631 (calc. 528.4619) and [M_4Na_–4Na]^4−^ ion peak at *m/z* 390.6005 (calc. 390.5991) in the (−) HR-ESI-MS (Appendix A). Chilensoside D (**5**), analogously to compounds **1**–**4**, has a pentasaccharide branched by C-4 Xyl1 chain consisting of xylose, quinovose, two glucose and 3-*O*-methylglucose residues deduced from thorough analysis of its 1D and 2D NMR spectra (Table 6, Appendix A). The availability of four-charged ion peaks in the ESI-MS spectra of **5** indicated four sulfate groups are present in its carbohydrate chain. The analysis of 1D TOCSY spectrum corresponding to Glc3 showed strongly deshielded signals of protons of the hydroxy methylene group at δ_H_ 4.61 (m) and 5.00 (d, *J* = 11.9 Hz), which were assigned to the corresponding carbon signal at δ_C_ 67.6. These data indicate that the glucose residue in the bottom semi-chain was sulfated by C-6. The glucose unit (Glc4) attached to C-4 Xyl1 in chilensoside D (**5**) had two sulfate groups at C-4 Glc4 and C-6 Glc4, deduced from the deshielding of its signals to δ_C_ 75.1 and 68.5, respectively. The fourth sulfate group was positioned at C-6 MeGlc5 because of the deshielding of the signals of hydroxy methylene group to δ_C_ 67.0 and δ_H_ 4.99 (brd, *J* = 11.9 Hz); 4.78 (dd, *J* = 5.1; 11.9 Hz). Therefore, chilensoside D (**5**) is a new, sixth tetrasulfated glycoside found in sea cucumbers [7,44]. 

The (*−*) ESI-MS/MS of chilensoside D (**5**) (Appendix A) demonstrated the fragmentation of [M_4Na_−Na]^−^ ion at *m/z* 1631.5, leading to the presence of the ion peak at *m/* 987.4 [M_4Na_−Na−MeGlcOSO_3_Na−Glc(OSO_3_Na)_2_]^−^, of [M_4Na_−2Na]^2−^ ion at *m/z* 804.2 leading to the ion peak at *m/z* 753.2 [M_4Na_−2Na−SO_3_Na+H]^2−^, and of [M_4Na_−3Na]^3−^ ion at *m/z* 528.5 leading to the ion peak at *m/z* 494.3 [M_4Na_−3Na−SO_3_Na+H]^3−^.

These data indicate that chilensoside D (**5**) is 3*β*-*O*-{6-*O*-sodium sulfate-*β*-D-glucopyranosyl-(1 → 4)-*β*-D-quinovopyranosyl-(1 → 2)-[6-O-sodium sulfate-3-*O*-methyl-*β*-D-glucopyranosyl-(1 → 3)-4,6-*O*-sodium disulfate-*β*-D-glucopyranosyl-(1 → 4)]-*β*-D-xylopyranosyl}-16-oxoholosta-9(11),24(25)-diene.

The structural peculiarities of the glycosides of *P. chilensis* showed similarity to the compounds of the representatives of the order Dendrochirotida, i.e., sea cucumbers of the species *Thyonidium kurilensis* and *Psolus chitonoides* (the same architecture of the carbohydrate chains), *Psolus fabricii* (attachment of sulfates to C-4 Glc4 and C-6 Glc4) and *Cladolabes schmeltzii* (the same aglycones). All these data significantly support the phylogenetic closeness of the order Molpadida to the order Dendrochirotida, rather than to the order Aspidochirotida (in accordance with the system of Pawson and Fell). This order is absent in the last revision of the system of the class Holothuroidea, and the families, which were part of it, are now included in the orders Holothuriida, Persiculida and Synallactida [46]. The obtained structural data are in good agreement with the phylogenetic study of Holothuroidea using a multi-gene approach, which showed poor support of Molpadida as a sister group to Synallactida but demonstrated the close relationship of Molpadida to Dendrochirotida [46].

### 2.2. Bioactivity of the Glycosides

Cytotoxic activity of chilensosides A–D (**1**–**5**) against human cells, including erythrocytes and cancer cell lines SH-SY5Y, HeLa, DLD-1, HL-60, and THP-1, was studied. The earlier tested chitonoidoside L [7] was used as the positive control (Table 7). 

The less active compounds in the series were chilensosides A (**1**) and D (**5**). The first of these substances has a hydroxyl group in the aglycone side chain, which is the cause of the decrease in its membranolytic activity [3]. In fact, its structural analog chilensoside A_1_ (**2**), with the same carbohydrate chain and aglycone without the OH-group, demonstrated high hemolytic and cytotoxic effects against all tested cell lines. Chilensoside D (**5**) is a tetrasulfated glycoside that itself is not the cause of activity depletion, because it is known that tetrasulfated hexaosides from *P. chitonoides*, chitonoidosides K and L, were significantly active [7]. The combination of sulfate group positions in **5**, especially at C-6 Glc3 and C-6 MeGlc5, probably negatively affected the activity. Disulfated chilensosides A_1_ (**2**), B (**3**) and trisulfated chilensoside C (**4**) displayed similar cytotoxicity. The differing sulfate groups in these glycosides were attached to C-4 or C-3 of glucopyranose units while C-6 positions of terminal sugar residues were free from sulfation.

The differential sensitivity of the cell lines in relation to the cytotoxic action of sea cucumber glycosides depended both on the glycoside’s chemical structures and the composition of cellular membranes [47]. In the current tests, erythrocytes were, as usual, more sensitive than cancer cells to the action of the glycosides, but leukemia cells (promyeloblast HL-60 and monocytic THP-1) displayed increased sensitivity compared to the other cancer cells.

Therefore, three of the five glycosides isolated from *P. chilensis* demonstrated high hemolytic and moderate cytotoxic activities against cancer cells. These data, along with the previous investigations of highly polar tri- and tetrasulfated glycosides [7], indicate the possible potential of these water-soluble compounds to be used as anticancer drugs.

## 3. Materials and Methods

### 3.1. General Experimental Procedures

Specific rotation, PerkinElmer 343 Polarimeter (PerkinElmer, Waltham, MA, USA); NMR, Bruker AMX 500 (Bruker BioSpin GmbH, Rheinstetten, Germany) (500.12/125.67 MHz (^1^Н/^13^C) spectrometer; Bruker AVANCE III-700 spectrometer at 700.13 MHz/176.04 MHz (1H/13C); ESI MS (positive and negative ion modes), Agilent 6510 Q-TOF apparatus (Agilent Technology, Santa Clara, CA, USA), sample concentration 0.01 mg/mL; HPLC, Agilent 1260 Infinity II with a differential refractometer (Agilent Technology, Santa Clara, CA, USA); columns Supelcosil LC-Si (4.6 *×* 150 mm, 5 µm) and Discovery HS F5-5 (10 *×* 250 mm, 5 µm) (Supelco, Bellefonte, PA, USA), Diasfer 110 C-8 (4.6 *×* 250 mm, 5 µm) (Biochemmack, Moscow, Russia).

### 3.2. Animals and Cells

Specimens of the sea cucumber *Paracaudina chilensis* (family Cuadinidae; order Molpadida) were collected in the Troitsa bay, Japan sea in August 2019 by scuba diving from 2–5 m depth. The animals were taxonomically determined by Boris B. Grebnev. Voucher specimens are kept in G.B. Elyakov PIBOC FEB RAS, Vladivostok, Russia.

Human erythrocytes were purchased from the Station of Blood Transfusion in Vladivostok. The cells of human adenocarcinoma line HeLa were provided by the N.N. Blokhin National Medicinal Research Center of Oncology of the Ministry of Health Care of the Russian Federation, (Moscow, Russia). The cells of human colorectal adenocarcinoma line DLD-1 CCL-221™, human promyeloblast cell line HL-60 CCL-240, human monocytic THP-1 TIB-202^™^ cells and human neuroblastoma line SH-SY5Y CRL-2266^™^ were received from ATCC (Manassas, VA, USA). HeLa cell line was cultured in DMEM (Gibco Dulbecco’s Modified Eagle Medium) with 1% penicillin/streptomycin sulfate (Biolot, St. Petersburg, Russia) and 10% fetal bovine serum (FBS) (Biolot, St. Petersburg, Russia). The cells of DLD-1, HL-60, and THP-1 lines were cultured in RPMI medium with 1% penicillin/streptomycin (Biolot, St. Petersburg, Russia) and 10% fetal bovine serum (FBS) (Biolot, St. Petersburg, Russia). All the cells were incubated at 37 °C in a humidified atmosphere with 5% (*v*/*v*) CO_2_. SH-SY5Y were cultured MEM (Minimum Essential Medium) with 1% penicillin /streptomycin sulfate (Biolot, St. Petersburg, Russia) and with fetal bovine serum (Biolot, St. Petersburg, Russia) to a final concentration of 10%. 

This study was conducted according to the guidelines of the Declaration of Helsinki and was approved by the Ethics Committee of the Pacific Institute of Bioorganic Chemistry (Protocol No. 0037.12.03.2021).

### 3.3. Extraction and Isolation

The sea cucumbers (36 specimens) were kept in EtOH at +4 °C. Then, they were minced by cutting into pieces and extracted with refluxing EtOH (2 L vol.) for 4 hrs. The extract was concentrated to dryness in vacuum, dissolved in H2O, and chromatographed on a Polychrom-1 column (powdered Teflon, Biolar, Latvia). We first eluted the inorganic salts and impurities with H2O and then the glycosides with 50% EtOH to give 1300 mg of crude glycoside fraction. This was subjected to column chromatography on Si gel using the stepwise gradient of solvent systems CHCl3/EtOH/H2O: 100:100:17 → 100:125:25 → 100:150:50 as mobile phase as the first stage of purification. Three fractions, I (377 mg), II (378 mg) and III (338 mg), were obtained. Each of them was subsequently rechromatographed on an Si gel column using the solvent system CHCl3/EtOH/H2O (100:125:25) as mobile phase, resulting in the isolation of subfractions: I.0 (22 mg), I.1 (120 mg), II (286 mg), III.1 (66 mg) and III.2 (177 mg). HPLC of the subfraction I.0 on silica-based column Supelcosil LC-Si (4.6 × 150 mm, 5 µm) with CHCl_3_/MeOH/H_2_O (55/30/4) as mobile phase resulted in the isolation of three fractions (I.0.1–I.0.3). The subsequent HPLC of fraction I.0.3 on Supelco Discovery HS F5-5 (10 × 250 mm) column with MeOH/H_2_O/NH_4_OAc (1 M water solution), ratio (50/48.5/1.5), as mobile phase led to the isolation of 5.5 mg of chilensoside A (**1**). HPLC of the subfraction I.1 on the silica-based column Supelcosil LC-Si (4.6 × 150 mm, 5 µm) in the same conditions used for the subtraction I.0 resulted in the isolation of three fractions (I.1.1–I.1.3). The HPLC of the fraction I.1.3 on Supelco Discovery HS F5-5 (10 × 250 mm) column with MeOH/H_2_O/NH_4_OAc (1 M water solution) (65/33.5/1.5) as mobile phase led to isolation of individual chilensosides A_1_ (**2**) (2.2 mg) and B (**3**) (4.8 mg) as well as the subfraction I.1.3.1. The repeated HPLC of the latter on the same column with MeOH/H_2_O/NH_4_OAc (1 M water solution) (55/43.5/1.5) as mobile phase resulted in the obtaining of 3.8 mg of chilensoside C (**4**). The HPLC of the subfraction III.1 on silica-based column Supelcosil LC-Si (4.6 × 150 mm, 5 µm) with CHCl_3_/MeOH/H_2_O (55/25/3) as mobile phase followed by HPLC of the obtained fraction on Diasfer 110 C-8 (4.6 × 250 mm) column with MeOH/H_2_O/NH_4_OAc (1 M water solution) (49/49/2) as mobile phase gave three fractions III.1.1–III.1.3. The rechromatography of III.1.3 on Diasfer 110 C-8 (4.6 × 250 mm) column with MeOH/H_2_O/NH_4_OAc (1 M water solution) (50/48/2) led to isolation of 2.2 mg of chilensoside D (**5**). 

#### 3.3.1. Chilensoside A (**1**)

Colorless powder; [α]_D_^20^*−*48° (*c* 0.1, 50% MeOH). NMR: See Table 1 and Table 2, Appendix A. (*−*)HR-ESI-MS *m/z*: 1443.4800 (calc. 1443.4815) [M_2Na_–Na]^−^, 710.2466 (calc. 710.2461) [M_2Na_–2Na]^2−^. (*−*)ESI-MS/MS *m/z*: 1179.5 [M_2Na_−Na−C_6_H_11_O_5_ (Glc)–SO_3_Na+2H]^−^, 1135.5 [M_2Na_−Na−C_6_H_11_O_5_ (Glc)–C_6_H_10_O_4_ (Qui)+H]^−^, 1010.4 [M_2Na_−Na−C_7_H_13_O_6_ (MeGlc)−2HSO_4_Na]^−^, 417.1 [M_2Na_−Na−C_7_H_13_O_6_ (MeGlc)−C_6_H_8_O_11_S_2_Na_2_ (Glc(OSO_3_Na)_2_)−C_30_H_43_O_4_ (Agl)]^−^, and 255.0 [M_2Na_−Na−C_7_H_13_O_6_ (MeGlc)−C_6_H_8_O_11_S_2_Na_2_ (Glc(OSO_3_Na)_2_)−C_6_H_11_O_5_ (Glc)−C_30_H_43_O_4_ (Agl)]^−^.

#### 3.3.2. Chilensoside A_1_ (**2**)

Colorless powder; [α]_D_^20^*−*36° (*c* 0.1, 50% MeOH). NMR: See Table 3 and Appendix A, Appendix A. (*−*)HR-ESI-MS *m/z*: 1427.4928 (calc. 1427.4865) [M_2Na_−Na]^−^, 702.2510 (calc. 702.2487) [M_2Na_–2Na]^2−^; (*−*)ESI-MS/MS *m/z*: 1119.5 [M_2Na_−Na−C_6_H_11_O_5_ (Glc)−C_6_H_10_O_4_ (Qui)+H]^−^, 915.4 [M_2Na_−Na−C_6_H_11_O_5_ (Glc)−C_6_H_10_O_4_ (Qui)−2SO_3_Na+3H]^−^, 667.1 [M_2Na_−Na−C_6_H_11_O_5_ (Glc)−C_6_H_10_O_4_ (Qui)–C_30_H_43_O_3_ (Agl)]^−^, 417.1 [M_2Na_−Na−C_7_H_13_O_6_ (MeGlc)−C_6_H_8_O_11_S_2_Na_2_ (Glc(OSO_3_Na)_2_)−C_30_H_43_O_3_ (Agl)]^−^.

#### 3.3.3. Chilensoside B (**3**)

Colorless powder; [α]_D_^20^*−*53° (*c* 0.1, 50% MeOH). NMR: See Table 4 and Appendix A, Appendix A. (*−*)HR-ESI-MS *m/z*: 1427.4881 (calc. 1427.4865) [M_2Na_–Na]^−^, 702.2499 (calc. 702.2487) [M_2Na_–2Na]^2−^; (*−*)ESI-MS/MS *m/z*: 1307.5 [M_2Na_−Na−NaHSO_4_]^−^, 1149.5 [M_2Na_−Na−C_7_H_12_O_8_SNa (MeGlcOSO_3_Na)+H]^−^, 987.4 [M_2Na_−Na−C_7_H_12_O_8_SNa (MeGlcOSO_3_Na)−C_6_H_11_O_5_ (Glc)+H]^−^, 841.4 [M_2Na_−Na−C_7_H_12_O_8_SNa (MeGlcOSO_3_Na)−C_6_H_11_O_5_ (Glc)−C_6_H_10_O_4_ (Qui)+H]^−^, 667.1 [M_2Na_−Na−C_30_H_43_O_3_ (Agl)−C_6_H_11_O_5_ (Glc)−C_6_H_10_O_4_ (Qui)−H]^−^, 621.7 [M_2Na_−2Na−C_6_H_11_O_5_ (Glc)]^2−^, and 548.2 [M_2Na_−2Na−C_6_H_11_O_5_ (Glc)−C_6_H_10_O_4_ (Qui)]^2−^

#### 3.3.4. Chilensoside C (**4**)

Colorless powder; [α]_D_^20^*−*56° (*c* 0.1, 50% MeOH). NMR: See Table 5 and Appendix A, Appendix A. (*−*)HR-ESI-MS *m/z*: 1529.4300 (calc. 1529.4253) [M_3Na_–Na]^−^, 753.2206 (calc. 753.2180) [M_3Na_–2Na]^2−^, 494.4839 (calc. 494.4823) [M_3Na_–3Na]^3−^; (*−*)ESI-MS/MS *m/z*: 1015.4 [M_3Na_−Na−C_6_H_10_O_8_SNa (GlcOSO_3_Na)−C_6_H_10_O_4_ (Qui)−SO_3_Na]^−^, 987.4 [M_3Na_−Na−C_7_H_13_O_5_ (MeGlc)−C_6_H_8_O_11_S_2_Na_2_ (Glc(OSO_3_Na)_2_)]^−^, 605.2 [M_3Na_−Na−C_7_H_13_O_5_ (MeGlc)−NaHSO_4_]^−^, 702.2 [M_3Na_−2Na−SO_3_Na]^2−^, 605.2 [M_3Na_−2Na−C_7_H_13_O_5_ (MeGlc)−NaHSO_4_]^2−^.

#### 3.3.5. Chilensoside D (**5**)

Colorless powder; [α]_D_^20^*−*61° (*c* 0.1, 50% MeOH). NMR: See Appendix A and Table 6, Appendix A. (*−*) HR-ESI-MS *m/z*: 1631.3667 (calc. 1631.3641) [M_4Na_–Na]^−^, 804.1886 (calc. 804.1874) [M_4Na_–2Na]^2−^, 528.4631 (calc. 528.4619) [M_4Na_–3Na]^3−^, 390.6005 (calc. 390.5991) [M_4Na_–4Na]^4−^ (Appendix A); (*−*) ESI-MS/MS *m/z* 987.4 [M_4Na_−Na−C_7_H_12_O_8_SNa (MeGlcOSO_3_Na)−C_6_H_7_O_11_S_2_Na_2_ (Glc(OSO_3_Na)_2_)]^−^, 753.2 [M_4Na_−2Na−SO_3_Na+H]^2−^, 494.3 [M_4Na_−3Na−SO_3_Na+H]^3−^. 

### 3.4. Cytotoxic Activity (MTT Assay) (for SH-SY5Y, HeLa and DLD-1 Cells)

All the studied substances (including chitonoidoside L used as positive control) were tested in concentrations between 0.1 µM to 100 µM using 2-fold dilution in d-H2O. The cell suspension (180 µL) and solutions (20 µL) of tested compounds in different concentrations were injected in wells of 96-well plates (SH-SY5Y, 1 × 104 cells/well, HeLa and DLD-1, 6 × 10^3^/200 µL) and incubated at 37 °C for 24 h in atmosphere with 5% CO_2_. Then, 100 µL of fresh medium was added instead of the tested substances in the same volume of medium. After that, 10 µL of MTT (3-(4,5-dimethylthiazol-2-yl)-2,5-diphenyltetrazolium bromide) (Sigma-Aldrich, St. Louis, MO, USA) stock solution (5 mg/mL) was added to each well, and the microplate was incubated for 4 h. Next, each well was additionally incubated for 18 h with 100 µL of SDS-HCl solution (1 g SDS/10 mL d-H2O/17 µL 6 N HCl). Multiskan FC microplate photometer (Thermo Fisher Scientific, Waltham, MA, USA) was used to measure the absorbance of the converted dye formazan at 570 nm. Cytotoxic activity of the tested compounds was calculated as the concentration that caused 50% cell metabolic activity inhibition (IC50). The experiments were carried out in triplicate, *p* < 0.05.

### 3.5. Cytotoxic Activity (MTS Assay) (for HL-60 and THP-1 Cells)

The cells of HL-60 line (10 × 10^3^/200 µL) and THP-1 (6 × 10^3^/200 µL) were placed in 96-well plates at 37 °C for 24 h in a 5% CO_2_ incubator. The cells were treated with tested substances and chitonoidoside L as positive control at concentrations from 0 to 100 µM for an additional 24 h incubation. Then, the cells were incubated with 10 µL MTS ([3-(4,5-dimethylthiazol-2-yl)-5-(3-carboxymethoxyphenyl)-2-(4-sulfophenyl)-2H-tetrazolium) for 4 h, and the absorbance in each well was measured at 490/630 nm with plate reader PHERA star FS (BMG Labtech, Ortenberg, Germany). The experiments were carried out in triplicate and the mean absorbance values were calculated. The results were presented as the percentage of inhibition that produced a reduction in absorbance after tested compounds treatment compared to the non-treated cells (negative control), *p* < 0.01.

### 3.6. Hemolytic Activity

Erythrocytes were isolated from human blood (AB(IV) Rh+) by centrifugation with phosphate-buffered saline (PBS) (pH 7.4) at 4 °C for 5 min by 450 g on centrifuge LABOFUGE 400R (Heraeus, Hanau, Germany) three times. Then, the residue of erythrocytes was resuspended in ice cold phosphate saline buffer (pH 7.4) to a final optical density of 1.5 at 700 nm, and kept on ice. For the hemolytic assay, 180 µL of erythrocyte suspension was mixed with 20 µL of test compound solution (including chitonoidoside L used as positive control) in V-bottom 96-well plates. After 1 h of incubation at 37 °C, plates were exposed to centrifugation for 10 min at 900 g on laboratory centrifuge LMC-3000 (Biosan, Riga, Latvia). Then, 100 µL of supernatant was carefully selected and transferred in new flat-plates, respectively. Lysis of erythrocytes was determined by measuring the concentration of hemoglobin in the supernatant with microplate photometer Multiskan FC (Thermo Fisher Scientific, Waltham, MA, USA), λ = 570 nm. The effective dose causing 50% hemolysis of erythrocytes (ED50) was calculated using the computer program SigmaPlot 10.0. All the experiments were carried out in triplicate, *p* < 0.01.

## 4. Conclusions

As a result of investigation of the glycosidic composition of the sea cucumber *Paracaudina chilensis,* the structures of five new glycosides, chilensosides A–D (**1**–**5**), were established and their cytotoxic activities were studied. Two different aglycones were found and one of them was a part of four compounds. Four diverse carbohydrate chains were detected in the studied glycosides. They differed in the quantity of sulfate groups: two in chilensosides of groups A (**1**, **2**) and B (**3**), three in chilensoside C (**4**), and four in chilensoside D (**5**). The positions of sulfation were also variable: two sulfates were attached to C-4 and C-6 of Glc4 residue in the glycosides **1**, **2**; additional third sulfate group bonded C-3 Glc3 in chilensoside C (**4**); two sulfates bonded different monosaccharide residues, C-6 Glc4 and C-4 Meglc5, in chilensoside B (**3**); and, finally, two positions of sulfation at C-6 Glc3 and C-6 MeGlc5, additional to those observed in **1**, **2**, were detected in chilensoside D (**5**). 

Such diversity in sulfate group quantity and positions indicates the high enzymatic activity of sulfatases. They have low specificity to attach a sulfate group to different positions of the same or several monosaccharide residues in glycosides **1**–**5**. Especially interesting was the observation that the sulfatase could compete with the glycosidase, bonding the sulfate group to C-3 Glc3 in chilensoside C (**4**) instead of potential glycosylation of this position. Additionally, it is interesting to note that only the aglycones with intranuclear 9(11)-double were found. This indicates that one oxidosqualene cyclase (OSC), forming the parkeol (precursor of the glycosides with 9(11)-double bond), is expressed in *P. chilensis*. 

The structures of the glycosides of *P. chilensis* were similar to those found in some representatives of the order Dendochirotida, confirming phylogenetic closeness of the order Molpadida to the order Dendrochirotida.

Rather high hemolytic and cytotoxic activity of three out of five isolated glycosides along with the previous investigations of highly polar tri- and tetrasulfated glycosides indicate the possible potential of these water-soluble compounds to be used as anticancer drugs. 

## Figures and Tables

**Figure 1 molecules-27-07655-f001:**
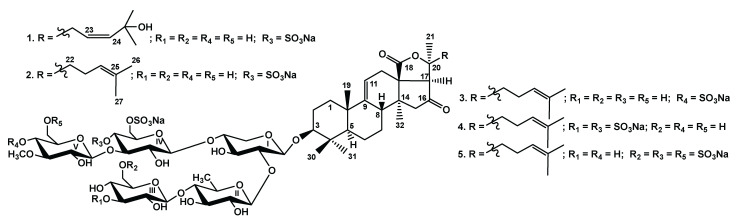
Chemical structures of glycosides isolated from *Paracaudina chilensis*: **1**—chilensoside A; **2**—chilensoside A_1_; **3**—chilensoside B; **4**—chilensoside C; **5**—chilensoside D.

**Table 1 molecules-27-07655-t001:** ^13^C and ^1^H NMR chemical shifts and HMBC and ROESY correlations of aglycone moiety of chilensoside A (**1**).

Position	δ_C_ Mult. *^a^*	δ_H_ Mult. (*J* in Hz) *^b^*	HMBC	ROESY
1	36.0 CH_2_	1.72 m		H-11
		1.31 m		H-3, H-5, H-11
2	26.6 CH_2_	2.06 m		
		1.84 m		H-30
3	88.6 CH	3.13 dd (4.5; 11.3)		H-5, H-31, H1-Xyl1
4	39.5 C			
5	52.7 CH	0.79 brd (11.2)	C: 10, 19	H-1, H-3, H-31
6	20.9 CH_2_	1.57 m		H-31
		1.38 m		H-30
7	28.2 CH_2_	1.56 m		
		1.17 m		H-32
8	38.6 CH	3.11 m		H-19
9	151.1 C			
10	39.6 C			
11	111.2 CH	5.26 brs		H-1
12	31.8 CH_2_	2.61 d (16.7)		H-17, H-32
		2.45 dd (6.7; 16.7)		H-21
13	55.8 C			
14	42.0 C			
15	52.0 CH_2_	2.41 d (15.2)	C: 13, 16	H-32
		2.15 d (15.2)		
16	214.6 C			
17	61.1 CH	2.89 s	C: 12, 16, 18	H-12, H-21, H-32
18	176.8 C			
19	21.9 CH_3_	1.26 s	C: 1, 5, 9, 10	H-1, H-2, H-8, H-30
20	82.8 C			
21	27.0 CH_3_	1.44 s	C: 17, 20, 22	H-12, H-17, H-22
22	41.9 CH_2_	2.55 dd (4.8; 13.8)		H-24
		2.33 m		H-24
23	123.9 CH	5.90 dd (6.3; 11.8)	C: 22	
24	140.1 CH	5.90 d (11.8)	C: 25	H-22, H-26, H-27
25	81.3 C			
26	24.7 CH_3_	1.42 s	C: 24, 27	H-24
27	24.7 CH_3_	1.42 s	C: 24, 26	H-24
30	16.6 CH_3_	0.96 s	C: 3, 4, 5, 31	H-2, H-6, H-19, H-31
31	27.9 CH_3_	1.13 s	C: 3, 4, 5, 30	H-3, H-5, H-6, H-30, H-1 Xyl1
32	20.5 CH_3_	0.88 s	C: 8, 13, 14, 15	H-7, H-12, H-15, H-17

*^a^* Recorded at 176.04 MHz in C_5_D_5_N/D_2_O (4/1). *^b^* Recorded at 700.13 MHz in C_5_D_5_N/D_2_O (4/1). The original spectra of **1** are provided in Appendix A.

**Table 2 molecules-27-07655-t002:** ^13^C and ^1^H NMR chemical shifts and HMBC and ROESY correlations of carbohydrate moiety of chilensoside A (**1**).

Atom	δ_C_ Mult. *^a,b c^*	δ_H_ Mult. (*J* in Hz) *^d^*	HMBC	ROESY
Xyl1 (1 → C-3)				
1	104.7 CH	4.66 d (7.2)	C: 3	H-3; H-3, 5 Xyl1
2	**82.0** CH	3.93 t (8.6)		H-1 Qui2
3	75.1 CH	4.13 t (8.6)	C: 4 Xyl1	H-1, 5 Xyl1
4	**78.3** CH	4.11 m		H-1 Glc4
5	63.5 CH_2_	4.32 m	C: 3 Xyl1	
		3.61 m		H-1 Xyl1
Qui2 (1 → 2Xyl1)				
1	104.4 CH	5.02 d (7.1)	C: 2 Xyl1	H-2 Xyl1; H-3, 5 Qui2
2	75.5 CH	3.88 t (9.1)	C: 1, 3 Qui2	H-4 Qui2
3	75.1 CH	4.02 t (9.1)	C: 2, 4 Qui2	H-1, 5 Qui2
4	**86.3** CH	3.51 t (9.1)	C: 3 Qui2, 1 Glc3	H-1 Glc3; H-2 Qui2
5	71.4 CH	3.70 dd (6.1; 9.2)		H-1, 3 Qui2
6	17.9 CH_3_	1.61 d (6.1)	C: 4, 5 Qui2	H-4 Qui2
Glc3 (1 → 4Qui2)				
1	104.6 CH	4.80 d (8.1)	C: 4 Qui2	H-4 Qui2; H-3, 5 Glc3
2	74.4 CH	3.86 t (8.7)	C: 1, 3 Glc3	
3	77.2 CH	4.13 m	C: 4 Glc3	H-1, 5 Glc3
4	70.9 CH	3.92 m	C: 5 Glc3	
5	77.6 CH	3.92 m	C: 6 Glc3	H-1, 3 Glc3
6	61.8 CH_2_	4.39 d (11.2)		H-4 Glc3
		4.06 m	C: 5 Glc3	
Glc4 (1 → 4Xyl1)				
1	102.2 CH	4.87 d (7.8)	C: 4 Xyl1	H-4 Xyl1; H-3, 5 Glc4
2	73.8 CH	3.93 t (8.9)	C: 1 Glc4	
3	**82.9** CH	4.37 t (8.9)	C: 2, 4 Glc4; 1 MeGlc5	H-1 MeGlc5; H-1 Glc4
4	*75.6* CH	4.78 t (8.9)	C: 3, 5, 6 Glc4	
5	74.5 CH	4.27 t (8.9)		H-1 Glc4
6	*68.2* CH_2_	5.49 m		
		4.71 dd (8.9; 11.2)		
MeGlc5 (1 → 3Glc4)				
1	104.4 CH	5.20 d (7.8)	C: 3 Glc4	H-3 Glc4; H-3,5 MeGlc5
2	74.3 CH	3.99 t (8.9)	C: 1, 3 MeGlc5	
3	86.9 CH	3.65 t (8.9)	C: 2, 4 MeGlc5; OMe	H-1, 5 Me Glc5; OMe
4	70.0 CH	3.91 t (8.9)	C: 3, 5 MeGlc5	
5	77.4 CH	3.86 m		H-1 MeGlc5
6	62.0 CH_2_	4.34 d (12.3)		H-4 MeGlc5
		4.09 dd (6.7; 12.3)	C: 5 MeGlc5	
OMe	60.5 CH_3_	3.76 s	C: 3 MeGlc5	

*^a^* Recorded at 176.04 MHz in C_5_D_5_N/D_2_O (4/1). *^b^* Bold = interglycosidic positions. *^c^* Italic = sulfate position. *^d^* Recorded at 700.13 MHz in C_5_D_5_N/D_2_O (4/1). Multiplicity by 1D TOCSY. The original spectra of **1** are provided in Appendix A.

**Table 3 molecules-27-07655-t003:** ^13^C and ^1^H NMR chemical shifts, HMBC and ROESY correlations of aglycone moiety of chilensoside A_1_ (**2**).

Position	δ_C_ Mult. *^a^*	δ_H_ Mult. (*J* in Hz) *^b^*	HMBC	ROESY
1	36.0 CH_2_	1.72 m		H-11
		1.31 m		
2	26.7 CH_2_	2.07 m		
		1.85 m		
3	88.6 CH	3.13 dd (5.0; 12.3)		H-5, H-31, H1-Xyl1
4	39.5 C			
5	52.7 CH	0.79 brd (11.2)		H-1, H-3, H-7, H-31
6	20.8 CH_2_	1.58 m		
		1.39 m		H-19, H-30
7	28.2 CH_2_	1.58 m		H-15
		1.16 m		
8	38.6 CH	3.13 m		H-19
9	151.0 C			
10	39.6 C			
11	111.2 CH	5.27 m		H-1
12	32.0 CH_2_	2.64 brd (15.5)	C: 13, 18	H-32
		2.48 brd (15.5)		H-21
13	55.9 C			
14	42.0 C			
15	51.9 CH_2_	2.39 d (16.4)	C: 13, 16	H-7, H-32
		2.09 m		
16	214.6 C			
17	61.3 CH	2.88 s	C: 16, 18, 21	H-12, H-21, H-32
18	176.7 C			
19	21.9 CH_3_	1.26 s	C: 1, 5, 9, 10	H-1, H-2, H-8, H-30
20	83.4 C			
21	26.6 CH_3_	1.47 s	C: 17, 20, 22	H-12, H-17, H-23
22	38.6 CH_2_	1.80 m		
		1.58 m		
23	22.9 CH_2_	2.27 m		H-21
		2.01 m		
24	123.8 CH	5.03 m	C: 27	H-26
25	132.2 C			
26	25.4 CH_3_	1.56 s	C: 24, 25, 27	H-24
27	17.5 CH_3_	1.52 s	C: 24, 25, 26	H-23
30	16.5 CH_3_	0.95 s	C: 3, 4, 5, 31	H-2, H-6, H-19, H-31
31	27.9 CH_3_	1.13 s	C: 3, 4, 5, 30	H-3, H-5, H-6, H-30, H-1 Xyl1
32	20.5 CH_3_	0.89 s	C: 8, 13, 14, 15	H-7, H-12, H-15, H-17

*^a^* Recorded at 125.67 MHz in C_5_D_5_N/D_2_O (4/1). *^b^* Recorded at 500.12 MHz in C_5_D_5_N/D_2_O (4/1). The original spectra of **2** are provided Appendix A.

**Table 4 molecules-27-07655-t004:** ^13^C and ^1^H NMR chemical shifts and HMBC and ROESY correlations of carbohydrate moiety of chilensoside B (**3**).

Atom	δ_C_ Mult. *^a b c^*	δ_H_ Mult. (*J* in Hz) *^d^*	HMBC	ROESY
Xyl1 (1 → C-3)				
1	104.7 CH	4.66 d (7.9)	C: 3	H-3; H-3, 5 Xyl1
2	**82.0** CH	3.95 t (7.9)	C: 1 Qui2; 1, 3 Xyl1	H-1 Qui2
3	75.0 CH	4.15 t (7.9)	C: 4 Xyl1	H-1, 5 Xyl1
4	**78.2** CH	4.14 m	C: 3 Xyl1	H-1 Glc4
5	63.4 CH_2_	4.37 dd (5.3; 11.8)	C: 1, 3 Xyl1	
		3.62 m		H-1 Xyl1
Qui2 (1 → 2Xyl1)				
1	104.5 CH	5.02 d (7.1)	C: 2 Xyl1	H-2 Xyl1; H-3, 5 Qui2
2	75.6 CH	3.88 t (8.9)	C: 1, 3 Qui2	H-4 Qui2
3	75.0 CH	4.00 t (8.9)	C: 2, 4 Qui2	H-1, 5 Qui2
4	**86.2** CH	3.52 t (8.9)	C: 1 Glc3; 3, 5 Qui2	H-1 Glc3; H-2 Qui2
5	71.4 CH	3.69 m		H-1, 3 Qui2
6	17.8 CH_3_	1.62 d (5.1)	C: 4, 5 Qui2	H-4 Qui2
Glc3 (1 → 4Qui2)				
1	104.4 CH	4.81 d (7.8)	C: 4 Qui2	H-4 Qui2; H-3, 5 Glc3
2	74.4 CH	3.87 t (8.6)	C: 1, 3 Glc3	
3	77.2 CH	4.13 t (8.6)	C: 2, 4 Glc3	H-1, 5 Glc3
4	70.8 CH	3.93 m	C: 3 Glc3	
5	77.6 CH	3.92 m		H-1, 3 Glc3
6	61.8 CH_2_	4.39 d (11.2)		
		4.06 dd (4.3; 11.2)	C: 5 Glc3	
Glc4 (1 → 4Xyl1)				
1	102.3 CH	4.89 d (8.1)	C: 4 Xyl1	H-4 Xyl1; H-3, 5 Glc4
2	73.2 CH	3.83 t (9.3)	C: 1, 3 Glc4	
3	**85.9** CH	4.17 t (9.3)	C: 1 MeGlc5; 2, 4 Glc4	H-1 MeGlc5; H-1 Glc4
4	68.9 CH	3.86 t (9.3)	C: 3, 5, 6 Glc4	H-6 Glc4
5	75.1 CH	4.05 t (9.3)		H-1 Glc4
6	*67.2* CH_2_	4.95 d (11.2)		
		4.65 brd (11.2)	C: 5 Glc4	
MeGlc5 (1 → 3Glc4)				
1	104.3 CH	5.18 d (7.5)	C: 3 Glc4	H-3 Glc4; H-3,5 MeGlc5
2	74.0 CH	3.86 t (9.3)	C: 1, 3 MeGlc5	H-4 MeGlc5
3	85.2 CH	3.71 t (9.3)	C: 2, 4 MeGlc5; OMe	H-1 Me Glc5; OMe
4	*76.1* CH	4.88 t (9.3)	C: 3, 5, 6 MeGlc5	H-2, 6 MeGlc5
5	76.4 CH	3.85 t (9.3)		H-1 MeGlc5
6	61.7 CH_2_	4.50 d (11.2)		
		4.33 dd (5.6; 11.2)		
OMe	60.6 CH_3_	3.93 s	C: 3 MeGlc5	

*^a^* Recorded at 176.04 MHz in C_5_D_5_N/D_2_O (4/1). *^b^* Bold = interglycosidic positions. *^c^* Italic = sulfate position. *^d^* Recorded at 700.13 MHz in C_5_D_5_N/D_2_O (4/1). Multiplicity by 1D TOCSY. The original spectra of **3** are provided in Appendix A.

**Table 5 molecules-27-07655-t005:** ^13^C and ^1^H NMR chemical shifts and HMBC and ROESY correlations of carbohydrate moiety of chilensoside C (**4**).

Atom	δ_C_ Mult. *^a b,c^*	δ_H_ Mult. (*J* in Hz) *^d^*	HMBC	ROESY
Xyl1 (1 → C-3)				
1	104.7 CH	4.65 d (6.5)	C: 3	H-3; H-3, 5 Xyl1
2	**82.3** CH	3.89 t (8.4)	C: 1 Qui2; 1, 3 Xyl1	H-1 Qui2
3	75.0 CH	4.08 m	C: 4 Xyl1	H-1 Xyl1
4	**78.9** CH	4.07 m		H-1 Glc4
5	63.4 CH_2_	4.31 m	C: 3 Xyl1	
		3.60 dd (9.3; 11.2)		H-1, 3 Xyl1
Qui2 (1 → 2Xyl1)				
1	104.6 CH	4.94 d (7.5)	C: 2 Xyl1	H-2 Xyl1; H-3, 5 Qui2
2	75.4 CH	3.89 t (9.3)	C: 1, 3 Qui2	H-4 Qui2
3	74.8 CH	4.04 t (9.3)	C: 2, 4 Qui2	H-1, 5 Qui2
4	**86.1** CH	3.51 t (9.3)	C: 1 Glc3; 3, 5 Qui2	H-1 Glc3; H-2 Qui2
5	71.5 CH	3.68 dd (6.5; 9.3)		H-1, 3 Qui2
6	17.7 CH_3_	1.60 d (6.5)	C: 4, 5 Qui2	H-4 Qui2
Glc3 (1 → 4Qui2)				
1	104.4 CH	4.82 d (8.5)	C: 4 Qui2	H-4 Qui2; H-3, 5 Glc3
2	73.1 CH	3.90 t (8.5)	C: 3 Glc3	
3	*84.3* CH	5.03 t (8.5)	C: 2, 4 Glc3	H-1, 5 Glc3
4	69.8 CH	3.93 m	C: 3, 5 Glc3	
5	77.1 CH	3.93 m		H-1, 3 Glc3
6	61.5 CH_2_	4.35 brd (11.7)		
		3.99 d (11.7)	C: 5 Glc3	
Glc4 (1 → 4Xyl1)				
1	102.4 CH	4.85 d (8.5)	C: 4 Xyl1	H-4 Xyl1; H-3, 5 Glc4
2	73.7 CH	3.93 t (8.5)	C: 1, 3 Glc4	H-4 Glc4
3	**82.9** CH	4.37 t (8.5)	C: 1 MeGlc5; 2, 4 Glc4	H-1 MeGlc5; H-1, 5 Glc4
4	*75.6* CH	4.78 t (8.5)	C: 3, 5, 6 Glc4	H-2 Glc4
5	74.3 CH	4.28 t (8.5)	C: 4 Glc4	H-1, 3 Glc4
6	*68.3* CH_2_	5.50 brd (9.0)		
		4.70 brd (9.9)	C: 5 Glc4	
MeGlc5 (1 → 3Glc4)				
1	104.2 CH	5.21 d (7.7)	C: 3 Glc4	H-3 Glc4; H-3,5 MeGlc5
2	74.6 CH	4.00 t (8.6)	C: 1, 3 MeGlc5	
3	86.9 CH	3.65 t (8.6)	C: 2, 4 MeGlc5; OMe	H-1, 5 Me Glc5; OMe
4	70.0 CH	3.91 t (8.6)	C: 3, 5, 6 MeGlc5	
5	77.4 CH	3.87 t (8.6)		H-1, 3 MeGlc5
6	62.0 CH_2_	4.35 brd (11.5)		
		4.09 dd (5.7; 11.5)	C: 5 MeGlc5	H-4 MeGlc5
OMe	60.3 CH_3_	3.76 s	C: 3 MeGlc5	

*^a^* Recorded at 176.04 MHz in C_5_D_5_N/D_2_O (4/1). *^b^* Bold = interglycosidic positions. *^c^* Italic = sulfate position. *^d^* Recorded at 700.13 MHz in C_5_D_5_N/D_2_O (4/1). Multiplicity by 1D TOCSY. The original spectra of **4** are provided in Appendix A.

**Table 6 molecules-27-07655-t006:** ^13^C and ^1^H NMR chemical shifts and HMBC and ROESY correlations of carbohydrate moiety of chilensoside D (**5**).

Atom	δ_C_ Mult. *^a,b,c^*	δ_H_ Mult. (*J* in Hz) *^d^*	HMBC	ROESY
Xyl1 (1 → C-3)				
1	104.7 CH	4.63 d (8.0)	C: 3	H-3; H-5 Xyl1
2	**82.7** CH	3.74 t (8.0)	C: 1 Qui2; 1, 3 Xyl1	H-1 Qui2
3	75.5 CH	4.01 t (8.0)	C: 2, 4 Xyl1	
4	**80.7** CH	3.97 t (8.0)		H-1 Glc4
5	63.5 CH_2_	4.40 dd (5.3; 11.5)		
		3.64 dd (8.0; 11.5)		H-1 Xyl1
Qui2 (1 → 2Xyl1)				
1	104.5 CH	4.77 d (8.6)	C: 2 Xyl1	H-2 Xyl1; H-3, 5 Qui2
2	75.2 CH	3.91 t (8.6)	C: 1, 3 Qui2	H-4 Qui2
3	74.4 CH	4.04 t (8.6)	C: 4 Qui2	H-1 Qui2
4	**86.1** CH	3.29 t (8.6)	C: 1 Glc3; 3 Qui2	H-1 Glc3; H-2 Qui2
5	71.6 CH	3.62 t (8.6)		H-1 Qui2
6	17.8 CH_3_	1.55 d (6.1)	C: 4, 5 Qui2	H-4 Qui2
Glc3 (1 → 4Qui2)				
1	104.6 CH	4.64 d (7.9)	C: 4 Qui2	H-4 Qui2; H-3 Glc3
2	73.9 CH	3.75 t (8.6)	C: 1, 3 Glc3	
3	76.8 CH	4.10 t (8.6)	C: 2, 4 Glc3	H-1 Glc3
4	70.6 CH	3.81 t (8.6)	C: 3, 5, 6 Glc3	
5	75.1 CH	4.07 m		H-1 Glc3
6	*67.6* CH_2_	5.00 d (11.9)		
		4.61 m		
Glc4 (1 → 4Xyl1)				
1	103.4 CH	4.80 d (7.4)	C: 4 Xyl1	H-4 Xyl1; H-3, 5 Glc4
2	73.3 CH	3.95 t (8.6)	C: 1, 3 Glc4	
3	**79.4** CH	4.50 t (8.6)	C: 1 MeGlc5; 2, 4 Glc4	H-1 MeGlc5; H-1, 5 Glc4
4	*75.1* CH	4.78 m		
5	73.8 CH	4.28 t (8.6)		H-1, 3 Glc4
6	*68.5* CH_2_	5.46 dd (7.5; 12.5)		
		4.66 m		
MeGlc5 (1 → 3Glc4)				
1	101.8 CH	5.33 d (7.6)	C: 3 Glc4	H-3 Glc4; H-3,5 MeGlc5
2	73.5 CH	4.00 t (7.6)	C: 1, 3 MeGlc5	
3	86.4 CH	3.62 t (7.6)	C: 4 MeGlc5; OMe	H-1, 5 Me Glc5; OMe
4	69.5 CH	4.01 t (7.6)	C: 5 MeGlc5	
5	75.7 CH	4.01 t (7.6)		H-1, 3 MeGlc5
6	*67.0* CH_2_	4.99 brd (11.9)		
		4.78 dd (5.1; 11.9)		
OMe	60.4 CH_3_	3.75 s	C: 3 MeGlc5	

*^a^* Recorded at 125.67 MHz in C_5_D_5_N/D_2_O (4/1). *^b^* Bold = interglycosidic positions. *^c^* Italic = sulfate position. *^d^* Recorded at 500.12 MHz in C_5_D_5_N/D_2_O (4/1). Multiplicity by 1D TOCSY. The original spectra of **5** are provided in Appendix A.

**Table 7 molecules-27-07655-t007:** The cytotoxic activities of glycosides **1**–**5**, and chitonoidoside L (positive control) against human erythrocytes, and SH-SY5Y, HeLa, DLD-1, HL-60, THP-1 human cell lines.

Glycosides	ED_50_, µM, Erythrocytes	Cytotoxicity, IC_50_ µM				
		SH-SY5Y	HeLa	DLD-1	HL-60	THP-1
Chilensoside A (**1**)	4.85 ± 0.10	>100.00	>100.00	>100.00	84.90 ± 2.96	75.29 ± 2.12
Chilensoside A_1_ (**2**)	1.45 ± 0.12	38.78 ± 0.08	21.80 ± 0.22	29.78 ± 2.34	22.78 ± 0.60	30.26 ± 1.60
Chilensoside B (**3**)	0.96 ± 0.01	30.73 ± 2.65	20.34 ± 0.32	45.08 ± 1.63	5.68 ± 0.05	19.57 ± 0.75
Chilensoside C (**4**)	1.18 ± 0.05	26.43 ± 0.91	17.79 ± 0.56	35.44 ± 1.76	14.21 ± 1.23	18.59 ± 0.79
Chilensoside D (**5**)	10.26 ± 0.53	>100.00	>100.00	>100.00	64.74 ± 3.63	56.23 ± 2.74
Chitonoidoside L	1.24 ± 0.02	7.58 ± 0.13	10.88 ± 0.04	11.48 ± 0.25	6.96 ± 0.44	9.00 ± 0.75

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
