# Peer review of "The Isolation, Structure Elucidation and Bioactivity Study of Chilensosides A, A1, B, C, and D, Holostane Triterpene Di-, Tri- and Tetrasulfated Pentaosides from the Sea Cucumber Paracaudina chilensis (Caudinidae, Molpadida)"

_molecules, 2022, doi:10.3390/molecules27217655_

Round 1

Reviewer 1 Report

This manuscript is suitable to be published in Molecules and I recommend its publication without any changes.

Author Response

We appreciate Reviewer for high estimation of our research.

Reviewer 2 Report

In this manuscript, Alexandra S. Silchenko et al. identified 5 interesting sulfated triterpene glycosides from Sea Cucumber Paracaudina chilensis and reported their cytotoxic activities. Somehow the chemical evidence supported the phylogenetic closeness. This lab has a good record of discovery for this family marine natural products and cited multiple their own publications. The data including the structure identification part are convincing. Some revisions/updates are discussed below and the manuscript should be published after that.

·        Remove "First" in the title.

·        L39-L43: Two dependent clauses and without independent clause are here, which is not correct in English writing.

·        L62: Please re-organize the ratio as uniform forma, so that the readers can recognize the gradients changing clearly.

·        In the Extraction and isolation part, please provide the retention time for all the isolates and specify the extraction method, e.g. sonication or homogenization.

·        L75-L76: Please check the m/z data with your raw data in SI.

·        Add figures with 2D correlations and the HR-ESI fragmentation patterns, which are common in natural products community and easier to interpret the structures than the values in table only.

·        Convert Table 7 to a bar-graph.

·        Further polishing the English in this paper is suggested.

Author Response

We are grateful to Reviewer for the comments, there are our replies:

  1. "First" in the title was removed.
  2. L43-L51: The paragraph was changed and, accordingly, the ordinal numbers of references [9–13] were changed to [17-21]: “The taxonomic status of the order Molpadida rises questions of biologists dealing with the systematics of Holothuroidea. Some of them supported the idea of closeness of Molpadida and Dendrochirotida, while the others considered molpadiids as being near to Aspidochirotida [17]. From this viewpoint, the investigations of the glycosides of representatives of the order Molpadida are relevant both for the searching for new structural variants, which broaden the knowledges concerning chemical biodiversity, and chemotaxonomy. The glycosides are successfully used as chemotaxonomic markers of different sea cucumber systematic groups [18–21]. So, the analysis of chemical peculiarities of the glycosides of chilensis can help to resolve the dilemma.
  3. L65: The phrase was corrected to: “Its subsequent separation by the chromatography on Si gel columns with the stepped gradient of the system of eluents CHCl3/EtOH/H2O used in ratios (100:100:17), (100:125:25), and (100:150:50)”
  4. Extraction and Isolation part: the specification was added “were minced by cutting into pieces”. It is senseless to provide the retention times (Rt) for the obtained compounds because each of them was isolated by multistage HPLC procedure with changing of columns and solvents. The adding of Rt for each stage will make text harder to read. Moreover, the identification of glycosides by Rt is impossible due to the variability of natural extracts.
  5. L80-L81, L390: Please check the m/z data with your raw data in SI: the inaccuracy in the text is fixed.
  6. Although, it is common for natural product papers to provide figures with 2D correlations and the HR-ESI fragmentation for easier interpretation, but in the case of the glycosides the overall 2D NMR spectra look uninterpretable without digital zoom in Topspin software which used for structure elucidation. So, its providing doesn’t alleviate the interpretation. Moreover, all kind of NMR spectra are provided in Supporting Materials. As regards the MS/MS data it is difficult to illustrate all pathways of fragmentations because some ions are formed as result of the loss of several functional groups or sugar residues simultaneously from different part of the molecule. So, we would like to keep traditional form of data presentation.
  7. The requirement of the Reviewer “Convert Table 7 to a bar-graph” is not fulfilled because it will make the data less informative than the accurate values of ED50 and IC50 presented in Table 7.
  8. The English editing was made.

Reviewer 3 Report

The main goal of present paper is isolation and structure determination for triterpene pentaglycosides isolated from marine cucumber Paracaudina chilensis. The sea organisms including also cucumbers has a long history in chemistry of natural products (Se-Kwon Kim, S.W.A. Himaya, Advances in Food and Nutrition Research, Volume 65, p.297-317, Elsevier 2012). Authors isolated five chemical individuals and made deep analysis of their structure. The structural analysis is a main part of submitted report. The obtained derivatives exhibit rather moderate anticancer activity in studied model cells. They have rather high haemolytic activity which cross out eventual applications in living systems. The references contain several earlier papers of authors 14 versus 19 of all references. Important publications are pass over, e.g.,

Muhammad Abdul Mojid Mondol, Hee Jae Shin, M. Aminur Rahman,Mohamad Tofazzal Islam, Mar. Drugs 2017, 15, 317-352.

Se-Kwon Kim, S.W.A. Himaya, Advances in Food and Nutrition Research, Volume 65, p.297-317, Elsevier 2012

Author Response

We fully agree with the Reviewer on the high level of self-citation in the manuscript, so the Introduction part was broadened and the corresponding references [8–15] were added: “The most part of recent researches concerning the sea cucumber triterpene glycosides deals with the structure elucidation of the compounds isolated from representatives of the orders Dendrochirotida, Synallactida, Holothuriida [1–9]. These studies provide significant input to the exploration of chemical diversity, functions [10, 11], biosynthesis [12, 13] and biological activity of huge collection of these natural products, which are the valuable source of new drug-candidates [14, 15].”

  1. Mondol, M.A.M.; Shin, H.J.; Rahman, M.A. Sea cucmber glycosides: chemical structures, producing species and important biological properties. Drugs 2017, 15, 317. doi:10.3390/md15100317
  2. Vien, L.T.; Hanh, T.T.H.; Quang, T.H.; Thanh, Q.N.V.; Thao, D.T.; Cuong, N.X.; Nam, N.H.; Thung, D.C.; Kiem, P.V. Triterpene tetraglycosides from Stichopus herrmanni Semper, 1868. Prod. Commun. 2022, 17, 5. doi:10.1177/1934578X221105369
  3. Kamyab, E.; Rohde, S.; Kellerman, M.Y.; Schupp, P.J. Chemical defense mechanisms and ecological implications of Indo-Pacific holothurians. Molecules 2020, 25, 4008, 2–25. doi: 10.3390/molecules25204808
  4. Park, J.-I.; Bae, H.-R.; Kim, Ch.G.; Stonik, V.A.; Kwak, J.Y. Relationships between chemical structures and functions of triterpene glycosides isolated from sea cucumbers. Chem. 2014, 2, 77. https://doi.org/10.3389/fchem.2014.00077
  5. Li, Y.; Wang, R.; Xun, X.; Wang, J.; Bao, L.; Thimmappa, R.; Ding, J.; Jiang, J.; Zhang, L.; Li, T.; Lv, J.; Mu, C.; Hu, X.; Zhang, L.; Liu, J.; Li, Y.; Yao, L.; Jiao, W.; Wang, Y.; Lian, S.; Zhao, Z.; Zhan, Y.; Huang, X.; Liao, H.; Wang, J.; Sun, H.; Mi, X.; Xia, Y.; Xing, Q.; Lu, W.; Osbourn, A.; Zhou, Z.; Chang, Y.; Bao, Z.; Wang, S. Sea cucumber genome provides insights into saponin biosynthesis and aestivation regulation. Cell Discov. 2018, 4, 29. https://doi.org/10.1038/s41421-018-0030-5
  6. Claereboudt, E.J.S.; Gualier, G.; Decroo, C.; Colson, E.; Gerbaux, P.; Claereboudt, M.R.; Schaller, H.; Flammang, P.; Deleu, M.; Eeckhaut, I. Triterpenoids in echinoderms: fundamental differences in diversity and biosynthetic pathways. Drugs 2019, 17, 352. doi: 10.390/md17060352.
  7. Kim, S.K.; Himaya, S.W.A. Triterpene glycosides from sea cucucmbers and their biological activities. Food Nutr. Res. 2012, 65, 297-317. https://doi.org/10.1016/B978-0-12-416003-3.00020-2
  8. Khotimchenko, Pharmacological potential of sea cucumbers. Int. J. Mol. Sci. 2020, 19, 1342. https://doi.org/10.3390/ijms19051342

The Reviewer is completely right in saying: “The obtained derivatives exhibit rather moderate anticancer activity in studied model cells. They have rather high haemolytic activity which cross out eventual applications in living systems”. The erythrocytes are traditional and convenient model for the investigation of membranolytic action of the glycosides, so they were used for the purpose to demonstrate the membranotropic action of the compounds. Moreover, it is known that the glycosides show immunomodulatory effect when applied in nanomolecular concentrations. So, high haemolytic activity (in micromolar doses) is not a direct restriction to their application in living systems. The testing of cytotoxic activities of the glycosides from P. chilensis against cancer cell lines is a part of extensive series of preliminary anticancer screening studies of these compounds in order to select the perspective candidates. Moreover, the isolated glycosides might be active against other model cell lines demonstrating the selectivity. That will have to be the subject of future studies.

Reviewer 4 Report

Overall a sound scientific analysis of the isolated Tetrasulfated Pentaosides. The paper, however, requires extensive editing of English language and style, particularly in abstract and introduction. Later the paper reads better, but punctuation is often missing making it difficult to read and understand the the text. I also felt that more references are needed (where possible) in further support of the specific assignments. Major revisions are required due to the above, but also because of the high self-citation rate and similarity in the text with respect to previously published works by the author. 

Author Response

The extensive editing of English language and style was made.

We fully agree with the Reviewer on the high level of self-citation in the manuscript, so the Introduction part is broadened and references to the papers of the other authors are added. The references supporting the specific assignments (NMR signals for structure elucidation) are correct because contain data concerning the same structural elements as discussed in the text. Moreover, the compared NMR spectra are registered using the same solvents and conditions.

Round 2

Reviewer 3 Report

Thank you considering my comments. I agree with authors arguments, that the haemolytic activity of investigated compounds does not cross out their application in therapy but reduces the therapeutic window to narrow range of concentration. Manuscript has deep structural analysis of holostane triterpene pentasaccharide derivatives. Structural analysis is well done and can be useful for other researchers working in natural products chemistry. In experimental part” Extraction and Isolation” please add the weight of used biological material and volume of EtOH. These values are important to understand the ratio of isolated extract to raw material.

Author Response

Thanks to the Reviewer for the second round of review of our manuscript. The answers to the comments are listed below:

In experimental part” Extraction and Isolation” please add the weight of used biological material and volume of EtOH. These values are important to understand the ratio of isolated extract to raw material.

Answers:

L358: The specification “36 specimens” was added to the text. The dry residue has not been weighted due to a lot of sand in the sample, that is explained by the digging behavior of P. chilensis, so the gut was full of sand.

L359: The volume of used EtOH was added (2L vol.)

Reviewer 4 Report

Overall, an improved second version that can be considered for publication.

I just point out that (although much improved) self-citation rate is still a bit high. Please check with editors/journal policy.

Also experimental section 3.4 starting at "After incubation..." is identical to https://www.ncbi.nlm.nih.gov/pmc/articles/PMC8708177/.  Please if possible rephrase or delete and give reference.

Author Response

We are gratefull to the Reviewer for the second roud of reviewing and providing the additional comments.

I just point out that (although much improved) self-citation rate is still a bit high. Please check with editors/journal policy.

Answer:

All appropriate references to the other authors investigating different aspects of sea cucumber secondary metabolites were added to the manuscript and only necessary self-citations reflecting the modern state of the field of research or describing the compounds with the same structural elements required for identification were cited.

Also, experimental section 3.4 starting at "After incubation..." is identical to https://www.ncbi.nlm.nih.gov/pmc/articles/PMC8708177/.  Please, if possible, rephrase or delete and give reference.

Answer:

L433-439: The methodology description was rewritten to avoid self-plagiarism: “Then, 100 µL of fresh medium were added instead of the tested substances in the same volume of medium. After that, 10 µL of MTT (3-(4,5-dimethylthiazol-2-yl)-2,5-diphenyltetrazolium bromide) (Sigma-Aldrich, St. Louis, MO, USA) stock solution (5 mg/mL) was added to each well, and the microplate was incubated for 4 h. Next, each well was additionally incubated for 18 h with 100 µL of SDS-HCl solution (1 g SDS/10 mL d-H2O/17 µL 6 N HCl). Multiskan FC microplate photometer (Thermo Fisher Scientific, Waltham, MA, USA) was used to measure the absorbance of the converted dye formazan at 570 nm.”